# Genome-Wide Identification of NAC Family Genes in Oat and Functional Characterization of *AsNAC109* in Abiotic Stress Tolerance

**DOI:** 10.3390/plants13071017

**Published:** 2024-04-03

**Authors:** Yahui Xu, Jialong Cheng, Haibin Hu, Lin Yan, Juqing Jia, Bin Wu

**Affiliations:** 1College of Agriculture, Shanxi Agricultural University, Jinzhong 030801, China; xuyahui0213@163.com (Y.X.); chengjialong0110@sina.com (J.C.); 2Institute of Crop Science, Chinese Academy of Agricultural Sciences, Beijing 100081, China; 3State Key Laboratory of Crop Gene Resources and Breeding, Beijing 100081, China

**Keywords:** oat, *NAC* gene family, gene duplication, expression patterns, abiotic stress

## Abstract

The plant-specific *NAC* gene family is one of the largest transcription factor families, participating in plant growth regulation and stress response. Despite extensive characterization in various plants, our knowledge of the *NAC* family in oat is lacking. Herein, we identified 333 *NAC* genes from the latest release of the common oat genome. We provide a comprehensive overview of the oat *NAC* gene family, covering gene structure, chromosomal localization, phylogenetic characteristics, conserved motif compositions, and gene duplications. *AsNAC* gene expression in different tissues and the response to various abiotic stresses were characterized using RT-qPCR. The main driver of oat *NAC* gene family expansion was identified as segmental duplication using collinearity analysis. In addition, the functions of *AsNAC109* in regulating abiotic stress tolerance in *Arabidopsis* were clarified. This is the first genome-wide investigation of the *NAC* gene family in cultivated oat, which provided a unique resource for subsequent research to elucidate the mechanisms responsible for oat stress tolerance and provides valuable clues for the improvement of stress resistance in cultivated oat.

## 1. Introduction

Transcription factors (TFs) are proteins which bind to specific promoter regions of target genes and regulate gene expression. Intrinsic cellular processes are regulated by the actions of TFs, including development and differentiation, and signal pathway-mediated responses of cells to external perturbation [1]. Genes of the no apical meristem, *Arabidopsis* transcription activation factor, and cup-shaped cotyledon (*NAC*) TF family genes are widely distributed in many types of land plant species. Generally, NACs consist of a nuclear localization signal, a highly conserved N-terminal DNA-binding domain, and a variable-length and sequence C-terminal transcriptional activation domain that interacts with other TFs and probably plays a role in distinct developmental processes [2]. Usually, the N-terminal region has about 160 residues of amino acids and can be further subdivided into five functionally diverse subdomains, termed A-E. Substructural domain A is associated with the formation of functional dimmers that are important to stabilize the protein structure, substructural domains B and E diversify the protein’s functions, and the C and D subdomains are associated with DNA binding [3,4].

In plants, NACs play a critical role in the regulation of many vital biological processes, such as shoot apical meristem formation, secondary wall formation, cell division, embryo and flower organ development, lateral root formation, hormone signaling, leaf senescence, and fruit ripening [5,6]. Moreover, many studies have shown that NACs exert vital functions in regulating plant stress tolerance [7]. In *Arabidopsis*, *ANAC019*, *ANAC055*, and *ANAC072* were found to be induced by drought, high salinity, abscisic acid, and their overexpression in transgenic plants resulted in significant improvements in drought tolerance [8]. Overexpression of *OsNAC6* and *OsSNAC2* in rice enhanced seedling tolerance to drought, salt, and cold stresses [9]. In *Arabidopsis*, grapevine *VvNAC13* overexpression enhanced salt and cold stress tolerance and increased sensitivity to abscisic acid (ABA) [10]. Transformation of the *StNAC053* gene from potato into *Arabidopsis* augmented the transgenic plants’ salt tolerance, increased antioxidant enzyme activities (superoxide dismutase (SOD), catalase (CAT), and peroxidase (POD)), and decreased ROS accumulation in *Arabidopsis* under high salt stress, thereby enhancing salt tolerance [11]. Silencing the expression of *CaNAC2* in chili peppers enhanced the plant’s sensitivity to low temperature, and the plants showed leaf wilting at an early stage under low-temperature stress [12]. Therefore, *NAC* family genes are important regulators of plant resistance to abiotic stress.

Oat (*Avena sativa* L.) is among the oldest cultivated crops and is grown widely across the world. In China, oat remains a staple human food in certain arid and saline regions of north China where staple crops such as corn and wheat cannot grow. In recent years, severe changes in the climate, particularly the increased frequency and intensity of droughts, have adversely affected oat yields. Considering the critical role of the *NAC* genes in stress tolerance, a better understanding of *NAC* genes in oat will be useful to improve the stress tolerance of oat. Since their first identification in petunia [13], *NAC* genes have been studied widely in various plants, including rice [14], wheat [15], and Tartary buckwheat [16]. However, the members of the *NAC* gene family associated with stress tolerance in oat are still largely unknown. The recently published oat reference genome sequence provides an opportunity to systematically characterize the oat *NAC* gene family. Herein, we present a comprehensive overview of the oat *NAC* gene family, gene structure, chromosome localization, phylogenetic characteristics, conserved motif compositions, the expression of *AsNAC* genes in different tissues, and the response to various abiotic stresses were analyzed comprehensively and systematically. In addition, the roles of one *AsNAC* gene, *AsNAC109*, in regulating abiotic stress tolerance in *Arabidopsis* were clarified. The identification of the oat *NAC* gene family will provide the basis for future function analysis to elucidate the mechanisms responsible for oat stress tolerance and represent valuable data to support molecular-assisted breeding in oat.

## 2. Results

### 2.1. Identification, Chromosomal Localization, and Physicochemical Characterization of NAC Family Members in the Oat Genome

Genome-wide analysis of the oat genome identified 333 *NAC* genes, which were named *AsNAC001*-*AsNAC333* in order of chromosomal position. The predicted properties of the oat *NAC* genes are shown in Appendix A, and the analyses showed that the features of the encoded *AsNAC* proteins varied widely. The 333 *AsNAC*s ranged from 157 to 747 aa in length, 17,940 to 85,938 Da in molecular weights, and 4.32 to 9.95 in isoelectric points. The 333 *NAC* genes were distributed on all 21 chromosomes (Figure 1), showing a heterogeneous distribution with a higher distribution on chromosomes Chr2A and Chr2D, with 23 and 28, respectively, and a minimum distribution of only 7 *NAC* genes on chromosomes Chr4C and Chr3D.

### 2.2. Phylogenetic and Collinearity Analysis of AsNAC Genes

In order to learn more about the evolutionary history of the *NAC* gene family and to determine the phylogenetic relationships of the *AsNAC* genes in oats, we built phylogenetic trees of 117 and 333 *NAC* genes from *Arabidopsis* and oats (Appendix A), respectively, using MEGA 11 software. The 333 *AsNAC*s were divided into 14 subgroups, with varying numbers of *AsNAC*s in each group. The largest branch was the *NAC3* subfamily, containing 59 *AsNAC*s, and the smallest branch was the *NAC3* subfamily, containing only 8 *AsNACs*.

New genes with similar or different functions can be expanded via gene duplication. Using the MCScan X program, a collinearity study of the *NAC* genes was carried out to find the duplicates within the *AsNAC* gene family. The results (Appendix A) showed that there were 242 pairs of duplicated genes among the 333 *AsNAC* genes, with a large number of them located on chromosome Chr 5C (Appendix A). To further understand the phylogeny of the *AsNAC* gene family and assess if the protein-coding genes are under selective pressure, we calculated the ratio of non-synonymous substitutions (Ka) to synonymous substitutions (Ks). The Ka/Ks analysis (Appendix A) revealed that 242 gene pairs had Ka/Ks < < 1, indicating that they were evolutionarily conserved, had experienced strong purifying selection, and that they were functionally conserved during evolution.

### 2.3. Analysis of Conserved Protein Motifs and Cis-Acting Elements of the AsNAC Genes

To probe the structural evolution of the AsNAC proteins, we analyzed the exon–intron structural patterns of their encoding genes as well as their conserved motifs. Ten conserved protein motifs, Motifs 1–10, with essentially identical distributions were detected in 333 AsNAC proteins. Most AsNACs contain Motif 1, and based on this, we believe that Motif 1 is a conserved motif. The majority of AsNAC proteins within the same subgroup displayed similar motif composition, despite the significant levels of variability between subgroups. For example, the conserved motifs of subclade b are essentially the same, most proteins contain six motifs, from the N-terminus to the C-terminus: Motif 1, Motif 9, Motif 8, Motif 6, Motif 7, Motif 2. Among them, we found that Motif 2 is present in most AsNACs but is specifically absent in subfamily c. Notably, most AsNAC proteins have several specific motifs (1, 5, 4, 3, 6, 2, 7) clustered at the N-terminus.

Their gene structures were analyzed, and there was a high degree of variability in the number of introns and exons, ranging from two to seven exons, although most *AsNAC* genes have three exons. In general, members of the same subgroup typically share exon/intron configurations and gene lengths. For example, subgroup b and subgroup m contain 1 exon, while subgroups c and d contain 2 or 3 exons (Appendix A).

To investigate the functions of the oat *NACs*, we predicted the promoter region of the *AsNAC* genes, using the 2000 bp upstream of the transcription start site as the expected promoter region of the *AsNAC* genes. Multiple phytohormone-responsive elements were identified in addition to light-responsive, stress-responsive, and anaerobic-responsive elements (Appendix A). These comprised 24 light-responsive elements, 7 plant growth and development-related responsive elements, 5 stress-responsive elements, and 10 phytohormone-responsive elements. The CAT-box element is predominantly present in genes that respond to plant growth. Among the stress and phytohormone response-related cis elements, antioxidant response elements (AREs) are essential for anaerobic induction of cis elements, MYB3-binding promoter motifs (MBSs) are associated with drought responses, and long terminal repeats (LTRs) are involved in stress responses, such as high and low temperatures, methylation inhibitors, irradiation, and high salt. ABA-responsive elements (ABREs) are associated with the response to ABA, CGTCA and TGACG motifs are associated with the response to methyl jasmonate (MeJA), and P-boxes are associated with the response to gibberellic acid (GA). These elements of the response class may enable the *AsNAC* genes to play an important role in abiotic stresses.

### 2.4. Expression Pattern Analysis of the AsNAC Genes

Oat *NAC* gene expression was analyzed in four tissue types and under seven stress treatments (Appendix A). Analysis of the results showed that there were significant variations in the expression of genes in the leaves, stems, spikes, and roots, with the highest expression found in the latter. The expression levels of the vast majority of *AsNAC* genes were upregulated under low-temperature, high-temperature, drought, drought high-temperature, salt, alkali, and saline stresses, with the most significant upregulation being observed under drought and high-temperature stress.

### 2.5. Functional Characterization of the Transcriptional Activation Activity of the AsNAC109 Gene

To determine the transactivation ability of *AsNAC109*, we performed a yeast one-hybrid assay. Full-length *AsNAC109* and different fragments were cloned into pGBKT7 vector to identify its ability to activate transcription of the marker *HIS3* gene. Defective media screening assay showed that yeast containing pGBKT7, pGBKT7-*AsNAC109*, pGBKT7-*AsNAC109*-N, pGBKT7-*AsNAC109*-M, and pGBKT7-*AsNAC109*-C plasmids could grow on SD/-Trp single-deficient medium (Figure 2). By contrast, yeast with pGBKT7, pGBKT7-*AsNAC109*-M, and pGBKT7-*AsNAC109*-C plasmids failed to grow on double-deficient and triple-deficient media, whereas yeast with full-length of *AsNAC109* and N-terminus were able to grow normally. These results demonstrated that the *AsNAC109* transcription factor possesses the ability to activate transcription, and that this activity is mostly concentrated at the N-terminus of the factor rather than the C-terminus.

### 2.6. Subcellular Localization of AsNAC109

To clarify the location of *AsNAC109* in cells, pCAMBIA1300-GFP empty vector and pCAMBIA1300-*AsNAC109*-GFP recombinant vector were transiently expressed in *N. benthamiana* young leaves. Confocal imaging showed that the control cells emitted green, fluorescent signals at both the cell membrane and the nucleus (Figure 3), demonstrating that the pCAMBIA1300-GFP fusion protein was expressed at the cell membrane and nucleus. The fusion protein AsNAC109-GFP mainly emitted green, fluorescent signals in the nucleus, suggesting that *AsNAC109* is mainly located in the nucleus, which is consistent with the characteristics of transcription factors.

### 2.7. Identification of the Phenotype of AsNAC109-Overexpressing Arabidopsis and Determination of Physiological Indices

To identify the function of the *AsNAC109* gene, the overexpression vector pCAMBIA1300 vector with a 35S promoter was constructed and transformed into *Arabidopsis*. After antibiotic screening and PCR validation, four homozygous transgenic lines over-expressing *AsNAC109* (OE) were selected for further analysis, and their *AsNAC109* expression levels showed differences using RT-qPCR, while the control line (WT) had no detectable *AsNAC109* transcript. Phenotypic characterization tests showed that transgenic and WT plants behaved significantly differently under stress conditions. In general, the WT plants were significantly smaller and the root system was shorter, and the leaves mostly appeared to be yellowish or white under salt, alkali, and saline stresses, compared with those in plants overexpressing *AsNAC109* (Figure 4). The primary root lengths of the four *AsNAC109*-overexpressing *Arabidopsis* lines were longer than those of the WT plants under different stress conditions, and they were significantly different from each other. In addition, the results of the preliminary resistance characterization of the *AsNAC109*-overexpressing *Arabidopsis* lines showed that they had stronger resistance to stress than the WT.

To further validate the resistance of *AsNAC109*-overexpressing *Arabidopsis* in soil and the resistance regulatory mechanism of the *AsNAC109* gene, resistance in soil and physiological indices under stress treatments were assessed (Figure 5). After 14 d of stress treatment, some of the leaves of WT *Arabidopsis thaliana* turned yellow, and the plants were dwarfed. After stress treatment of *AsNAC109*-overexpressing *Arabidopsis*, the leaf tips of the old leaves turned yellow, but the inner young leaves remained green, and overall, the leaves did not show symptoms of dryness, the degree of stress injury was less than that of the WT, and their growth was significantly better than that of the WT. The *AsNAC109*-overexpressing *Arabidopsis* lines exhibited reduced MDA levels and increased POD and SOD enzyme activity in comparison with the WT. The higher SOD activities of the *AsNAC109*-overexpressing *Arabidopsis* lines indicated that the transgenic *Arabidopsis* had an enhanced ability to scavenge ROS. The accumulation of MDA leads to membrane lipid peroxidation, which impairs the membrane structure, thus impacting a series of physiological and biochemical processes. The significantly decreased accumulation of MDA in the transgenic *Arabidopsis* suggested that it suffered a lower degree of cellular damage under adverse stress.

The ionic concentrations of Na^+^, Ca^2+^, K^+^, and Cl^−^ ions were examined in salt- and alkali-stressed transgenic *Arabidopsis* and WT plants (Figure 6). The results showed that the concentrations of Na^+^ and Cl^−^ in the transgenic *Arabidopsis* were lower than those in the WT under salt and alkali stress, while the concentrations of Ca^2+^ and K^+^ were higher in the transgenic *Arabidopsis* than in the WT. This suggested that the *AsNAC109*-overexpressing *Arabidopsis* possessed stronger ROS scavenging ability, which reduced their Na^+^ toxicity and oxidative damage under salt and alkali stresses, leading to enhanced salt and alkali tolerance.

## 3. Discussion

The important regulatory role of NAC TFs in plant growth and their potential value for utilization in plant improvement have led to *NAC* genes being identified at a genome-wide level in various plant species [17]. However, the oat genome is large and complex; therefore, we know little about the oat *NAC* gene family. Herein, based on the recently published cultivated oat genome, we performed a genome-wide identification of NAC domain TF genes in cultivated oat. Systematic analysis of gene structure, duplication events, phylogenetic relationships, and expression patterns in various tissues and in response to abiotic stresses were conducted. Moreover, the roles of one *AsNAC* gene, *AsNAC109*, in regulating abiotic stress tolerance in yeast and *Arabidopsis* were identified. The results of this study will further our understanding of *NAC* genes and help to elucidate the evolution and possible functions of NAC domain proteins in cultivated oat.

A previous study showed that the number of *NAC* genes diverged markedly among 160 tested species, with *Brassica napus* having the highest number (410) and *Marchantia polymorpha* having the lowest (9) [17]. In this study, a genome-wide search identified 333 NAC genes in cultivated oat, which is higher than that in *Arabidopsis* (115), rice (151) [18], and soybean (152) [19] but lower than that in bread wheat (488) [15], oil-seed rape (410) [17], and alfalfa (421) [20]. Compared with that in other plant species, the number of *NAC* genes identified in cultivated oat is relatively high. In most cases, differences in the number of *NAC* genes could be attributed to differences in the genome size of each species [21]. Therefore, the high number of *NAC* genes in oat is partly caused by the derivation of the cultivated hexaploid oat from the hybridization of three diploid ancestors, thus containing six sets of chromosomes (AA, CC, and DD subgenomes). Gene expansion is mostly caused by gene duplication events, including tandem, segmental, and whole-genome duplication [22]. Collinearity analysis showed that there are 157 duplication events among oat *NAC* genes, comprising 146 segmental duplication events and 11 tandem duplication events, indicating that segmental duplication dominated oat gene expansion events. Similar results were reported for *Miscanthus sinensis* [21] and *Triticum dicoccoides* [23]. Gene duplication can be driven by DNA break repair. In bread wheat, massive numbers of small-scale interchromosomal gene duplications took place in the genome following the repair of double-strand DNA breaks [15]. Analysis of the oat genome sequence showed a mosaic-like genome architecture resulting from frequent genomic rearrangements, which might also have contributed to *NAC* gene family expansion. Gene duplication is a fundamental process in evolution, promoting genetic redundancy and robustness, giving rise to new genes and functionalities by providing new nucleotide sequences and contributing to the adaptation to climate change over time [24]. Therefore, the study of gene duplication events can help to understand gene function and the molecular processes behind the development of some particular features.

Considering the hexaploidy of cultivated oat, we expected that three homologous copies of each gene would be present in each of the three subgenomes, resulting in homologous groups. In this study, analysis of *NAC* genes on each chromosome showed that the *NAC* genes of cultivated oat are distributed unevenly among the chromosomes, similar to the identified *NAC* genes in other plants, such as rice [14], wheat [15], and soybean [19]. The numbers of oat *NAC* genes identified per chromosome varies from 7 to 28. For most *NACs*, the number of each *NAC* gene identified per homolog varies frequently, indicating unequal preservation of homologous *NAC* genes. The number of *NAC* genes identified on each oat subgenome group also varied, with the highest number on subgenome A (118) and the lowest on subgenome C (105), which is different from wheat, in which Borrill et al. [15] found that most of the *NAC* genes retained their corresponding copies in all three subgenomes of wheat. Analysis of the oat genome sequence showed that, unlike wheat, the oat subgenome exhibits an unbalanced number of genes, specifically, the C subgenome appears to have 12% fewer genes than the A or D subgenomes [25], which is consistent with our findings.

Typically, prokaryotic genes are intronless; however, eukaryotes also contain a proportion of intronless genes. Of the 333 identified *AsNAC* genes, about 13% (44) are intronless, which is lower than the proportion of intronless *NAC* genes in wheat (19%). A lack of introns is a feature of retrotransposons, which “copy and paste” the DNA of the retrotransposon to a new location in the genome, leading to the expansion of certain genes or genetic elements in a genome [26]. Compared with oat, wheat has more intronless *NAC* genes, suggesting more retrotransposition events in the wheat genome. Moreover, we found that the distribution of *AsNAC* intronless genes on each chromosome did not correlate with the distribution of the whole *AsNAC* gene family. For example, although chromosome 7D has 19 *NAC* genes, which is higher than average, no intronless *NAC* genes were observed on chromosome 7D. Whereas chromosome 3D, which contains only seven *NAC* genes, has one intronless *NAC* gene. Chromosome 1A, which has a similar number of *NAC* genes (20) to Chromosome 7D, has the highest number intronless *NAC* genes (10). This divergence of *NAC* genes suggests that several mutations or loss of sequences might have occurred in specific homologs of the *NAC* gene family. Furthermore, most of the intronless *AsNAC* genes are classified into homologous groups, which suggested that retrotransposition events likely occurred before the polyploidization of oat.

The *NAC* TF gene family is one of the largest families identified to date [23]. Although the *NAC* gene structure is relatively conserved, the lengths, predicted molecular masses of the encoded proteins, and their pI, vary among *NAC* genes. Based on the phylogenetic analysis of *NAC* gene sequences from oat and the model plant *Arabidopsis*, we further divided the *AsNAC* gene family into 14 distinct clades comprising *NAC* genes with highly similar structures. Analysis of gene expression patterns can provide clues to protein function. Consequently, we used RT-qPCR to determine the *NAC* gene expression profiles in various tissues and environments. The results of RT-qPCR showed that, in general, for different tissues, *NAC* genes were highly expressed in roots and under drought and high-temperature conditions. The TERN subgroup, which contains h and k motifs, demonstrated higher root expression than the other subgroups. Interestingly, most members of the TERN subfamily, such as *AsNAC117/187/186/071/316/287/177/081*, also showed high levels of expression under different stress treatments. Roots are an important organ of plants to perceive harmful changes in the external environment [27]. Thus, *AsNAC* genes showing marked root-specific expression might exert vital functions in the response to abiotic stress. NAC TFs in the TERN subgroup contain a conserved WKATGSPG sequence, similar to that in WRKY proteins, suggesting a DNA-binding function for this conserved domain and the existence of a common evolutionary ancestor of the NAC-TF and WRKY TF families [28]. In cotton, TERN subgroup member *GbNAC1* is highly expressed in roots and stems. Analysis of the overexpression of *GbNAC1* in *Arabidopsis* treated with different stresses suggested that *GbNAC1* has an important function in abiotic stress resistance [29].

In a phylogenetic tree, genes on the same branch are likely to have similar functions and expression profiles [23,30]. In this study, we found that genes in the same clade with detected expression exhibited dissimilar expression profiles. For example, unlike other TERN subgroup members, the expression of *AsNAC023* was not induced by the stress conditions adopted in this study, although it was highly expressed in roots, which suggested that it might only function in root development. All members of TIP subgroup exhibited higher expression under salt stress; however, only *NAC058* also showed high expression under drought stress, the expression of other members did not change significantly, and the expression levels of *AsNAC307/091/196/267* were downregulated. Similar results were reported in mung bean and wheat [15,31]. Interestingly, tandem duplicated genes also exhibited dissimilar expression profiles. *AsNAC187*, a tandem duplicated gene of *AsNAC188*, showed high expression in the roots and the ears of grain, while *AsNAC188* did not. Moreover, cold, salt, and alkali stress induced *AsNAC187* expression, while *AsNAC188* expression was inhibited by the three stresses. This difference between tandem duplicated genes indicates that they might have undergone divergent evolution and acquired new functions, leading to the development of distinct adaptations suited to various environmental conditions. This suggests that although *NAC* genes evolved from a common ancestor, they underwent multiple replication events during differentiation and species formation, which resulted in divergent functions during plant growth and development. Moreover, analysis of the selection pressures on the *NAC* genes of cultivated oats showed that most tandem duplicated genes showed significant overall purifying selection (Ka/Ks < < 1), suggesting that purifying selection acted as the primary force, and some of the retained duplicated genes are still functional and might have an adaptive advantage.

Plant responses to abiotic stress, e.g., salinity, cold, drought, and heat, involve NAC TFs. However, we lack knowledge regarding the involvement of NAC TFs in the oat abiotic stress response. Previously, we obtained a new *AsNAC* gene, designated *AsNAC109*, which shows differential expression under stress; however, its function was largely unknown. To further clarify the roles of *AsNAC109* under abiotic stress, we genetically transformed *AsNAC109* into *Arabidopsis*. Investigation of phenotypic and stress-related physiological characteristics showed that stress tolerance was significantly enhanced in the transgenic plants compared with those in the WT. The length of the roots was longer, and the Ca^2+^/K^+^ contents were higher in the transgenic plants than in the WT plants under stress, suggesting that the *AsNAC109*-overexpressing plants suffered lower ion toxicity under salt and alkali stresses relative to the WT plants, and were better able to maintain ion homeostasis in vivo. Compared with WT, ROS accumulation in the transgenic plants was lower, and antioxidant enzyme activity was higher, indicating that *AsNAC109* might enhance stress tolerance by promoting ROS scavenging. The transcription of a wide variety of genes might be regulated by NAC TFs, resulting in enhanced tolerance to various stresses. Similar results have been obtained in other studies of *NAC* genes. In rice, knockout of *OsNAC45*, which belongs to the NAM subgroup like *AsNAC109*, resulted in more ROS accumulation and increased sensitivity of rice to salt stress [32]. In *Tamarix hispida*, *ThNAC4* overexpression in transgenic *T. hispida* and *Arabidopsis* enhanced antioxidant enzyme activities (SOD, POD, and GST) and the contents of osmoprotectants (proline and trehalose) under stress conditions [33]. Wheat *TaNAC2* overexpression in *Arabidopsis* resulted in enhanced resistance to drought, salt, and cold stresses, as demonstrated by elevated expression of the abiotic stress-responsive gene and other physiological indices [34]. Taken together, these findings imply that *AsNAC109* is a positive regulator of stress tolerance and may have a significant impact on it.

## 4. Materials and Methods

### 4.1. Growth Conditions and Plant Materials

Seeds of the common oat cultivar ‘*HanYou-5*’ were provided by The Institute of Crop Sciences, Chinese Academy of Agricultural Sciences, Beijing, China. The seeds were sterilized and incubated in germination boxes in a 16 h/8 h (light/dark) greenhouse at 28/25 °C and 70% relative humidity. Samples were obtained from four different parts of the leaves, stems, and roots of oat seedlings and the spikes of oats at the tasseling stage and placed at −80 °C. Oat seedlings grown for 7 d were subjected to drought (20% PEG6000), salt (150 mmol/L NaCl), alkali (100 mmol/L Na_2_CO_3_ + NaHCO_3_), saline (100 mmol/L NaCl, Na_2_SO_4_ + NaHCO_3_), cold (4 °C), and heat (42 °C) stress treatments. The roots, stems, and leaves of oat seedlings and spikes at the heading stage were harvested and placed at −80 °C until the extraction of total RNA. Oat seeds were sterilized and placed in a germination box and grown hydroponically, protected from light, for 10 days, and their leaves were then collected to prepare protoplasts [35].

Wild-type (WT) and T2 generation transgenic *Arabidopsis* seeds were sterilized and sown on normal 1/2 Murashige and Skoog (MS) medium for 4 d before stress treatment. Seedlings with uniform growth were selected and transplanted into different stress media, including drought (125 mmol/L PEG6000), salt (100 mmol/L NaCl), alkali (50 mmol/L Na_2_CO_3_+NaHCO_3_), and saline (100 mmol/L NaCl, Na_2_SO_4_ + NaHCO_3_), and six plants of each of WT and transgenic *Arabidopsis*, with three biological replicates, were treated. Photographs were taken, and root lengths were measured after vertical incubation for 7 d. For physiological measurement, transgenic *Arabidopsis* and WT control plants were grown in 1/2 MS medium for about 7 d, and individuals with a consistent growth state were selected and transplanted into pots with a matrix of Metromix 350 (Sungro Horticultural, Agawam, MA, USA). Three weeks after transplantation, the plants were subjected to the different stress treatments mentioned above, and the leaves were collected to determine the physiological indexes.

### 4.2. Genome-Wide Identification of Oat AsNAC Genes and Prediction of Physicochemical Properties of the Encoded Proteins

The oat and *Arabidopsis* genome sequence and annotation information were down-loaded from the NCBI database (https://www.ncbi.nlm.nih.gov/datasets/genome/GCA_916181665.1/ accessed on 2 February 2022 and https://www.ncbi.nlm.nih.gov/datasets/genome/GCF_000001735.4/ accessed on 15 March 2022). The NAC-conserved structural domain sequence (PF01849) was obtained from the PFAM database (http://pfam.xfam.org/ accessed on 13 April 2022). NAC family members were identified using HMMER 3.0, and the results were filtered based on a score value of greater than 90 and an E-value of less than 1 × 10^−5^ [36]. The NAC homologous sequences in oats were then identified based on *Arabidopsis* NAC protein sequences. The intersections of the PFAM and BLAST screening results were identified and validated using the Conserved Domain Database (CDD) database (https://www.ncbi.nlm.nih.gov/cdd/ accessed on 20 June 2022) to finally screen the oat NAC family genes. Using EXPASY, physicochemical characteristics for the AsNAC family members were predicted, including molecular weight and isoelectric point. (https://web.expasy.org/protparam/ accessed on 24 August 2022) [37].

### 4.3. Chromosomal Localization, Collinearity Analysis, and Phylogeny Analysis of the NAC Gene Family in Oat

Based on the oat genomic data, the length and chromosomal position of each found *AsNAC* gene were calculated and displayed using MapChart 2.32 [38]. The Multiple Collinearity Scan toolbox (MCScanX http://chibba.pgml.uga.edu/mcscan2/#tm accessed on 2 December 2022) was employed to analyze the tandem repeats in *AsNAC* genes [39], and the *AsNAC* gene duplicates were identified by Tbtools 2.0 [40].

The NAC protein sequences of the identified pairs from *Arabidopsis* and oat were compared using the Clustalx tool (http://www.clustal.org/clustal2/ accessed on 15 December 2022) [41]. The NJ method in MEGA-X was then used to create an unrooted phylogenetic tree with bootstrap support from 1000 replications [42]. The evolutionary tree was plotted using iTOL (https://itol.embl.de/ accessed on 25 December 2022).

### 4.4. Gene Structure and Analysis of Conserved Motifs and Cis-Acting Elements of the AsNAC Gene Family

Gene Structure Display Server 2.0, an online application, was used to evaluate the *AsNAC* gene structures (GSDS, http://gsds.cbi.pku.edu.cn/ accessed on 12 March 2023). AsNAC protein-conserved motifs were searched using MEME for Motif Elicitation (MEME) (http://meme-suite.org/tools/meme accessed on 28 March 2023; number of conserved motifs = 10, otherwise the system default parameters were used). In order to identify cis-acting elements, the promoter regions of the *AsNAC* genes were taken from the oat genome sequence and submitted to PlantCARE (https://bioinformatics.psb.ugent.be/ accessed on 18 May 2023) [43]. The results were collated and visualized using TBtools 2.0.

### 4.5. Expression Pattern Analysis of Oat NAC Gene Family Members Using Quantitative Real-Time Reverse-Transcription PCR (RT-qPCR)

RT-qPCR was carried out as described previously, and the internal housekeeping control gene was *EF1A* (encoding Elongation factor 1α) [44]. The RT-qPCR primers were designed using Primer Express and are listed in Appendix A. An examination of the sodium dodecyl sulfate (SDS) dissociation curve was used to confirm the amplification’s specificity. Melting curve analysis was employed to verify that each pair of primers produced a distinct amplicon. Gene expression levels were analyzed using the 2^−ΔΔCt^ method [45]. Expression data were visualized as heatmaps using TBtools 2.0 software.

### 4.6. AsNAC109 Transcriptional Self-Activating Activity and Subcellular Localization

*AsNAC109* was identified in oat in response to salt stress [46]. To identify the transcriptional activity of the *AsNAC109* gene, the full-length gene, DNA encoding the N-terminus (1–139 aa), DNA encoding the middle region (140–438 aa), and DNA encoding the C-terminus (440–775 aa) were ligated into vector pGBKT7 separately, while empty pGBKT7 comprised the negative control. The recombinant plasmids and the empty vector were transformed into AH109 yeast cells, and the positive transformants were first screened on single-deficient medium (SD/-Trp), followed by lineage culture on SD/-Trp, SD/-Trp/-His, and SD/-Trp/-His/-Ade medium [47]. For subcellular localization, *AsNAC109* was ligated into pCAMBIA1300 vector with a 35S promoter and GFP tag, and control (35S:GFP) was then transformed into *Agrobacterium tumefaciens* strain GV1301 using the freeze–thaw method. Transient expression of 35S: AsNAC109-GFP in tobacco (*Nicotiana benthamiana*) leaves was performed by agroinfiltration [48]. After agroinfiltration, the fluorescent signal was recorded with a confocal laser scanning microscope (LSM980, Carl Zeiss Microscopy GmbH., Jena, Thüringen, Germany).

### 4.7. Functional Analysis of AsNAC109

The *AsNAC109* ORF was amplified using primers listed in Appendix A and ligated into pCAMBIA1300 vector with a 35S promoter. The constructed overexpression vector was transformed into Agrobacterium GV1301 as above and then transformed into Arabidopsis Col-0 by the Agrobacterium tumefaciens-mediated floral dip method. The transformants were selected on 50 mgL^−1^ of kanamycin and further verified by PCR. Homozygous T3 progeny was used for the phenotypic analysis and RT-qPCR, as is shown above.

The length of the primary roots was measured for *AsNAC109*-overexpressing transformants (OE1, OE2, OE3, and OE4) and the WT after 7 d of vertical incubation on stress medium and MS medium. For physiological index measurement, normally grown and stress-treated *Arabidopsis* leaves were collected, and their malondialdehyde (MDA) content was measured using the thiobarbituric acid (TAB) method [49]. Peroxidase (POD) was measured by the method of [50]. A previously published method was used to determine the superoxide dismutase (SOD) activity [51]. Na^+^, Ca^2+^, and K^+^ contents were measured using flame photometry [52], and the Cl^−^ content was measured using a colorimetric method [53].

## 5. Conclusions

Herein, the oat *NAC* gene family was subjected to genome-wide analysis based on the recently published cultivated oat genome. The identified 333 *AsNAC* genes were classified into 14 subgroups. Their chromosomal distribution, structure, gene duplications, conserved motifs, phylogenetic relationships, and expression patterns in various tissues and under abiotic stresses were further investigated. Collinearity analysis identified that the main driver of *NAC* gene expansion was segmental duplication. Duplicated genes might undergo divergent evolution and exhibit dissimilar expression profiles, leading to the development of distinct adaptations suited to various environmental conditions. Additionally, we identified a new stress-responsive *NAC* gene, *AsNAC109*. Overexpression of *AsNAC109* increased antioxidant enzyme activities, and reduced ionic toxicity and oxidative damage, which significantly improved the stress resistance of transgenic plants. These findings provide new insights into the regulatory functions of *NAC* genes in the underlying mechanisms of stress tolerance in cultivated oat. Our findings also provide valuable candidate gene resources for genetic engineering and molecular breeding to improve oat abiotic stress tolerance.

## Figures and Tables

**Figure 1 plants-13-01017-f001:**
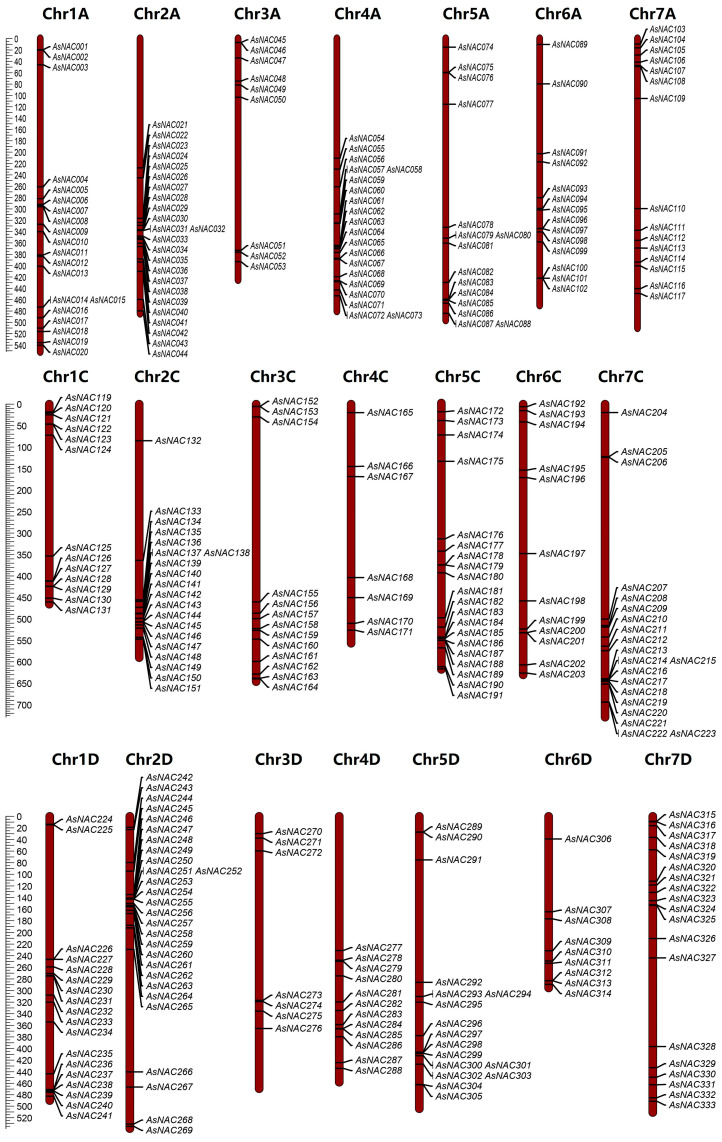
Oat *NAC* family genes’ chromosomal distribution. The chromosomes are represented using vertical bars, with the chromosome number appearing at the top. Chromosome length (Mb) is indicated by the scale appearing on the left.

**Figure 2 plants-13-01017-f002:**
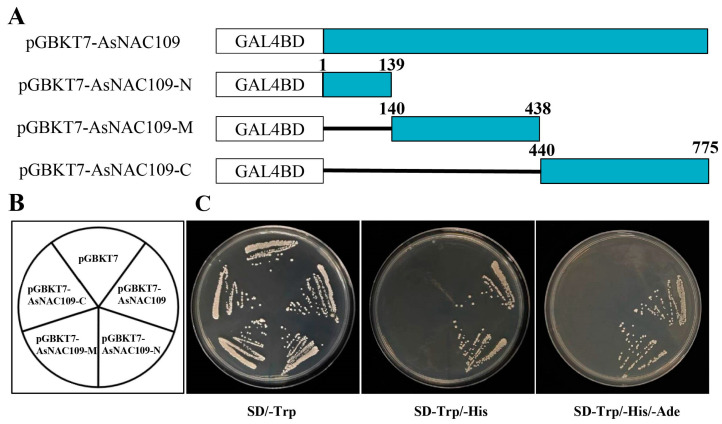
Transactivation AsNAC109 assayed in yeast. (**A**) Structures of the effectors employed in this experiment. In vector pGBKT7, the DNA-binding domain of GAL4 was fused to the full-length ORF, the N-terminal region, or the C-terminal region of AsNAC109. (**B**) Growth on selective media of AH109 cells harboring each of the constructs, the empty vector (negative control) or the positive control. (**C**) Transactivation was tested according to growth on SD-Trp, SD-Trp-His, and SD-Trp-His-Ade.

**Figure 3 plants-13-01017-f003:**
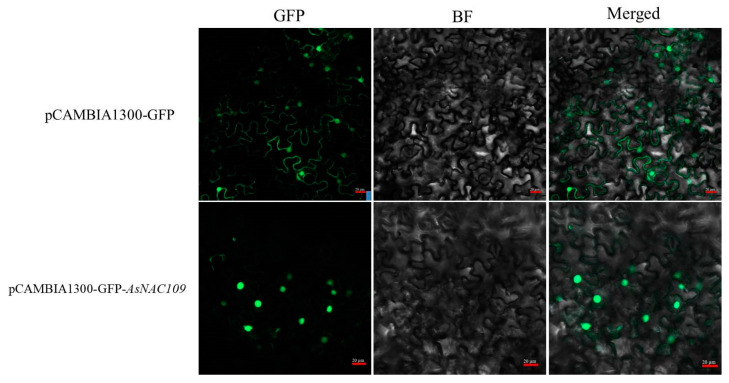
Subcellular localization of *AsNAC109*.bar = 20 μm; GFP: bright field, BF: green fluorescence, Merged: superimposed field, photographs were taken in three fields of view.

**Figure 4 plants-13-01017-f004:**
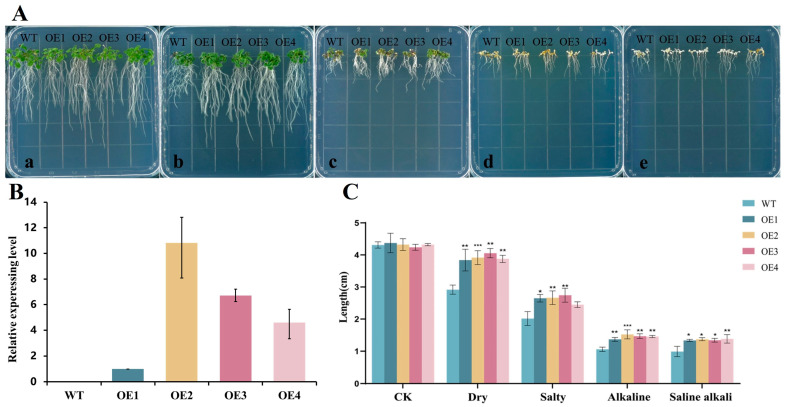
The root lengths of *AsNAC109*-overexpressing *Arabidopsis* lines under drought, salt, alkali, and saline stresses. (**A**) (a) Phenotypes of plants grown in MS medium; (b) phenotypes of plants grown in mannitol medium for 7 d; (c) phenotypes of plants grown in NaCl medium for 7 d; (d) phenotypes of plants grown in Na_2_CO_3_ and NaHCO_3_ (1:1) medium for 7 d; (e) phenotypes of plants grown in NaCl, Na_2_SO_4_, and NaHCO_3_ (1:1:1) medium for 7 d (WT represents wild-type Arabidopsis, OE1, OE2, OE3, and OE4 represent four *AsNAC109*-overexpressing *Arabidopsis* lines). (**B**) Identification of *AsNAC109* transgenic lines by RT-qPCR. The relative expression levels of the over-expression (OE) lines were determined by dividing their expression levels by that of line 1 which had the lowest expression level. (**C**) Mean primary root length of *Arabidopsis* plants under different stress treatments. (* marks significance at a level of 0.05, ** marks significance at a level of 0.01, and *** marks significance at a level of 0.001).

**Figure 5 plants-13-01017-f005:**
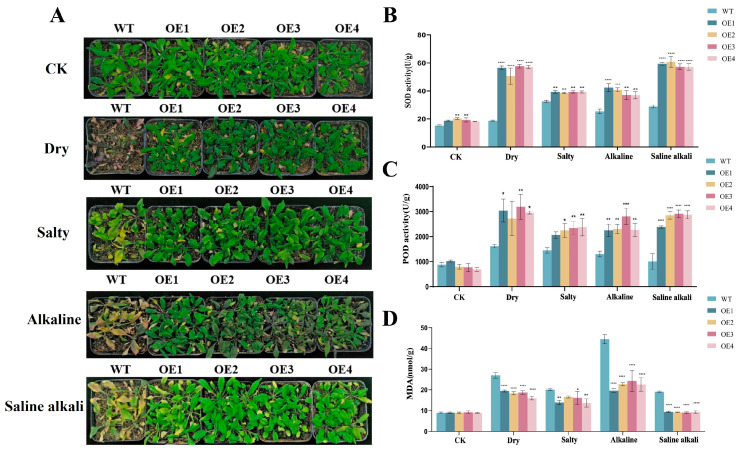
Phenotypic changes and the identification of physiological indicators of *AsNAC109* overexpression in *Arabidopsis thaliana* under stress conditions. (**A**) The growth of *Arabidopsis* under normal conditions, drought stress, salt stress, alkali stress, and saline and alkaline stress, respectively. (**B**) Activity of superoxide dismutase (SOD). (**C**) Activity of peroxidase (POD). (**D**) Malondialdehyde (MDA) content. OE1, OE2, OE3, and OE4 represent different *AsNAC109*-overexpressing *Arabidopsis* lines. * indicates a significant correlation at the 0.05 level, ** significant correlation at the 0.01 level, *** significant correlation at the 0.001 level, **** significant correlation at the 0.0001 level.

**Figure 6 plants-13-01017-f006:**
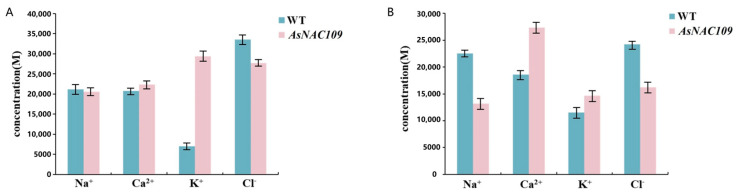
Ion concentrations under salt and alkali stress. (**A**) Ion concentrations of wild-type and *AsNAC109*-overexpressing *Arabidopsis* plants under salt stress. (**B**) Ion concentrations of wild-type and *AsNAC109*-overexpressing *Arabidopsis* plants under alkali stress.

## Data Availability

The data presented in this study are available on request from the corresponding author.

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
