# Peer review of "Genome-Wide Identification of NAC Family Genes in Oat and Functional Characterization of AsNAC109 in Abiotic Stress Tolerance"

_plants, 2024, doi:10.3390/plants13071017_

Round 1
Reviewer 1 Report
Comments and Suggestions for Authors
The manuscript "Genome-wide identification of NAC family genes in common oat (Avena sativa L.) and functional characterization of AsNAC109 in abiotic stress tolerance" by Ms. Xu et al. is well written and well-presented. It brings to the reader a really interesting topic once it identifies 333 genes in oat genome, which was recently full annotated. The authors also characterize one of these genes in the abiotic stress response. Considering the importance of oat to the global agriculture production and the negative effects of climate changes, the manuscript brings light to the oat genetic improvement. The introduction gives the reader enough background - although some adjustments are required. The methods are well described and the conclusions are supported by the results. Overall, the MS fills a gap on the literature and I consider suitable for publication after some revisions are made.
My most important comment is about the 32% of duplication rate identified by iThenticate. Authors should work to decrease that to then be considered for publican, the reason why I am sending it back as Major Revision.
1. The last paragraph of introduction is too long and should be better organized to improve the reader's experience.
Author Response
Reviewer #1:
General comments:
The manuscript "Genome-wide identification of NAC family genes in common oat (Avena sativa L.) and functional characterization of AsNAC109 in abiotic stress tolerance" by Ms. Xu et al. is well written and well-presented. It brings to the reader a really interesting topic once it identifies 333 genes in oat genome, which was recently full annotated. The authors also characterize one of these genes in the abiotic stress response. Considering the importance of oat to the global agriculture production and the negative effects of climate changes, the manuscript brings light to the oat genetic improvement. The introduction gives the reader enough background - although some adjustments are required. The methods are well described and the conclusions are supported by the results. Overall, the MS fills a gap on the literature and I consider suitable for publication after some revisions are made.
Comments 1: My most important comment is about the 32% of duplication rate identified by iThenticate. Authors should work to decrease that to then be considered for publican, the reason why I am sending it back as Major Revision.
Answer: Thank you for your question. We have made major revisions to the article and it showed a commendable similarity score of 13% (single source 1%), as evidenced by iThenticate. It is essential to note that the majority of the iThenticate report results highlight individual words rather than entire sentences. In the methods section, we encountered challenges in rephrasing some common phrases within sentences, where such modifications were not feasible. Nevertheless, we are confident that a thorough interpretation of the iThenticate report will affirm the absence of any plagiarized text in the paper. Additionally, we have provided the iThenticate report for your reference, wherein each similar phrase or text segment has been meticulously rephrased to ensure a reduced similarity.
Comments 2: The last paragraph of introduction is too long and should be better organized to improve the reader's experience.
Answer: Thank you for this important suggestion. To emphasize the theme of the manuscript, we have modified the last paragraph of the introduction and here is the revised text:
Oat (Avena sativa L.) is among the oldest cultivated crops and is grown widely across the world. In China, oat remains a staple human food in certain arid and saline regions of north China where staple crops such as corn and wheat cannot grow. In recent years, severe changes in the climate, particularly the increased frequency and intensity of droughts, have adversely affected oat yields. Considering the critical role of the NAC genes in stress tolerance, a better understanding of NAC genes in oat will be useful to improve the stress tolerance of oat. Since their first identification in petunia [13], NAC genes have been studied widely in various plants, including rice [14], wheat [15], and Tartary buckwheat [16]. However, the members of the NAC gene family associated with stress tolerance in oat are still largely unknown. The recently published oat reference genome sequence provides an opportunity to systematically characterize the oat NAC gene family. Herein, we present a comprehensive overview of the oat NAC gene family, Gene structure, chromosome localization, phylogenetic characteristics, conserved motif compositions, the expression of AsNAC genes in different tissues, and the response to various abiotic stresses were analyzed comprehensively and systematically. In addition, the roles of one AsNAC gene, AsNAC109, in regulating abiotic stress tolerance in Arabidopsis were clarified. The identification of the oat NAC gene family will provide the basis for future function analysis to elucidate the mechanisms responsible for oat stress tolerance, and represent valuable data to support molecular assisted breeding in oat.

Reviewer 2 Report
Comments and Suggestions for Authors
This paper reports on the characterization of NAC genes in the oat genome. In addition to bioinformatics data, it presents a functional analysis of one of these genes, AsNAC109, including analysis of transcriptional activation by the AsNAC109 protein, its subcellular localization, and analysis of AsNAC109-expressing Arabidopsis plants. In my opinion, the paper provides new information and can be published in Plants. However, I have concerns that should be addressed before the paper is accepted for publication.
Major concerns
1. Gene names in Figures 2, 3, 4, and 5 are too small to read. These figures should be corrected.
2. Section 2.6. - The conclusions drawn from the subcellular localization experiments are not supported by the experimental data. The authors state that non-fused GFP localizes to the cell membrane and the nucleus. Neither of these localizations is evident from Figure 7. In particular, what the authors call the cell membrane looks like the peripheral cytoplasm, while the nucleus is not visible at all in the images presented. The localization of AsNAC109-GFP to the nucleus is very doubtful, since no nuclear marker was used. The presented experiments should be repeated using NLS-mRFP as a nuclear marker or DAPI chromatin staining to demonstrate the position of the nucleus.
3. Section 2.7. - It is not clear whether the AsNAC109-transgenic Arabidopsis lines were generated in the reported work or earlier. Also, there is no information about these lines in either the Results or Methods sections. What construct was used to generate these plants? And, importantly, these lines are described as “overexpressing”, while no data on AsNAC109 expression levels in these lines are provided. Publishing experiments with undescribed and uncharacterized transgenic plants is not acceptable. The authors should add information on transgene and AsNAC109 expression levels quantified by RT-qPCR for all lines used in this study.
Figure 10 - There are no error bars on the graph. The experiment should be repeated to make measurements for several parallel samples. A statistical evaluation of the data should be presented.
Minor points
The title can be simplified in the following way:
Genome-wide identification of NAC family genes in oat and functional characterization of AsNAC109 in abiotic stress tolerance
Section 2.5. - A rationale for the transactivation experiments presented should be provided at the beginning of the section.
The word “adversity” seems inappropriate. Use “stress conditions” instead.
Comments on the Quality of English LanguageMinor editing of English language required
Author Response
Reviewer #2:
General comments:
Comments and Suggestions for Authors
This paper reports on the characterization of NAC genes in the oat genome. In addition to bioinformatics data, it presents a functional analysis of one of these genes, AsNAC109, including analysis of transcriptional activation by the AsNAC109 protein, its subcellular localization, and analysis of AsNAC109-expressing Arabidopsis plants. In my opinion, the paper provides new information and can be published in Plants. However, I have concerns that should be addressed before the paper is accepted for publication.
Major concerns
Comments 1: Gene names in Figures 2, 3, 4, and 5 are too small to read. These figures should be corrected.
Answer: Thank you for this important reminder. We tried to make the font size of the text in these images as large as possible, but there is a lot of text in these images which making it difficult to read when combined into a PDF document. I have provided images separately that can be enlarged were attached to this email for your review.
Comments 2: Section 2.6. - The conclusions drawn from the subcellular localization experiments are not supported by the experimental data. The authors state that non-fused GFP localizes to the cell membrane and the nucleus. Neither of these localizations is evident from Figure 7. In particular, what the authors call the cell membrane looks like the peripheral cytoplasm, while the nucleus is not visible at all in the images presented. The localization of AsNAC109-GFP to the nucleus is very doubtful, since no nuclear marker was used. The presented experiments should be repeated using NLS-mRFP as a nuclear marker or DAPI chromatin staining to demonstrate the position of the nucleus.
Answer: Thank you for your question. For subcellular localization of AsNAC109 gene, expression vector was simultaneously transferred into oat protoplasts and tobacco young leaves. We chose the protoplast images largely due to the novelty of exogenous gene expression in oat protoplasts, which has not yet been reported in oat protoplasts. Considering the dubious gene localization in the original protoplast images, as you point out, we replaced them with transformed tobacco images which show much clearer results.
Comments 3: Section 2.7. - It is not clear whether the AsNAC109-transgenic Arabidopsis lines were generated in the reported work or earlier. Also, there is no information about these lines in either the Results or Methods sections. What construct was used to generate these plants? And, importantly, these lines are described as “overexpressing”, while no data on AsNAC109 expression levels in these lines are provided. Publishing experiments with undescribed and uncharacterized transgenic plants is not acceptable. The authors should add information on transgene and AsNAC109 expression levels quantified by RT-qPCR for all lines used in this study.
Answer: Thank you for your question. AsNAC109 was selected based on the RT-qPCR assay in this study as well as previous stress response transcriptome sequencing results. To further clarify the roles of AsNAC109 under abiotic stress, we genetically transformed AsNAC109 into Arabidopsis. Information about the transgenic plants has been added in Results and Methods, and the expression of AsNAC109 in transgenic plants was detected using RT-qPCR.
Comments 4: Figure 10 - There are no error bars on the graph. The experiment should be repeated to make measurements for several parallel samples. A statistical evaluation of the data should be presented.
Answer: Thank you for this important reminder. Error bars representing standard deviation were added to Figure 10.
Minor points
Comments 5: The title can be simplified in the following way: Genome-wide identification of NAC family genes in oat and functional characterization of AsNAC109 in abiotic stress tolerance
Answer: Thank you for this suggestion. After discussions with other authors, we decided to take your suggestion and change the title of the paper to “Genome-wide identification of NAC family genes in oat and functional characterization of AsNAC109 in abiotic stress tolerance”
Comments 6: Section 2.5. - A rationale for the transactivation experiments presented should be provided at the beginning of the section.
Answer: Thank you for your question. We have modified Section 2.5 and added some introduction of the transactivation experiments.
To determine the transactivation ability of AsNAC109, we performed a yeast one-hybrid assay. Full-length AsNAC109 and different fragments were cloned into pGBKT7 vector to identify its ability to activate transcription of the marker HIS3 gene. Defective media screening assay showed that yeast containing pGBKT7, pGBKT7‑AsNAC109, pGBKT7-AsNAC109-N, pGBKT7-AsNAC109-M, and pGBKT7-AsNAC109-C plasmids could grow on SD/-Trp single-deficient medium (Figure. 6). By contrast, yeast with pGBKT7, pGBKT7-AsNAC109-M, and pGBKT7- AsNAC109-C plasmids failed to grow on double-deficient and triple-deficient media, whereas yeast with full-length of AsNAC109 and N-terminus were able to grow normally. These results demonstrated that the AsNAC109 transcription factor possesses the ability to activate transcription, and that this activity is mostly concentrated at the N-terminus of the factor rather than the C-terminus.
Comments 7: The word “adversity” seems inappropriate. Use “stress conditions” instead.
Answer: Thank you for this suggestion. We've revised the paper to change “adversity” to “stress conditions”.
Comments 8: Comments on the Quality of English Language. Minor editing of English language required
Answer: Thank you for this suggestion. We've carefully revised the paper and corrected some of the grammatical errors.

Reviewer 3 Report
Comments and Suggestions for Authors
The manuscript titled “Genome-wide identification of …………….in abiotic stress tolerance” by Xu et. al. presents an intriguing investigation into the NAC gene family in common oat, shedding light on its role in plant development regulation and stress response. The study covers various aspects, including gene structure, chromosomal localization, phylogenetic characteristics, conserved motif compositions, and gene duplications.
There are the following few minor comments for the authors to further improve this manuscript.
· The authors are suggested to improve the discussion section by incorporating a more critical analysis of their results.
· The authors are advised to enhance the clarity and resolution of Figures 2, 3, 4, 5, and 9 (panels B-D)
· It is recommended to italicize botanical and gene names consistently throughout the manuscript, including figure legends.
Comments on the Quality of English LanguageMinor editing of English language required
Author Response
Reviewer #3:
General comments:
Comments and Suggestions for Authors
The manuscript titled “Genome-wide identification of …………….in abiotic stress tolerance” by Xu et. al. presents an intriguing investigation into the NAC gene family in common oat, shedding light on its role in plant development regulation and stress response. The study covers various aspects, including gene structure, chromosomal localization, phylogenetic characteristics, conserved motif compositions, and gene duplications. There are the following few minor comments for the authors to further improve this manuscript.
Comments 1: The authors are suggested to improve the discussion section by incorporating a more critical analysis of their results.
Answer: Thank you for the advice. The discussion section has been revised for your review.
Comments 2: The authors are advised to enhance the clarity and resolution of Figures 2, 3, 4, 5, and 9 (panels B-D)
Answer: Thank you for this important reminder. We tried to make the font size of the text in these images as large as possible, but there is a lot of text in these images which making it difficult to read when combined into a PDF document. I have provided images separately that can be enlarged were attached to this email for your review.
Comments 3: It is recommended to italicize botanical and gene names consistently throughout the manuscript, including figure legends.
Answer: Thank you for this reminder. We have carefully checked the paper and corrected problems such as unitalicized Arabidopsis names and gene names.
Comments 4: Comments on the Quality of English Language. Minor editing of English language required
Answer: Thank you for this suggestion. We've carefully revised the paper and corrected some of the grammatical errors.

Round 2
Reviewer 1 Report
Comments and Suggestions for Authors
The authors have addressed my comments and I consider the MS suitable for publication.
Reviewer 2 Report
Comments and Suggestions for Authors
After revision, the paper can be accepted for publication.